# Research on a Face Real-time Tracking Algorithm Based on Particle Filter Multi-Feature Fusion

**DOI:** 10.3390/s19051245

**Published:** 2019-03-12

**Authors:** Tao Wang, Wen Wang, Hui Liu, Tianping Li

**Affiliations:** 1Shandong Key Laboratory of Medical Physics and Image Processing, School of Physics and Electronics, Shandong Normal University, Jinan 250014, China; wwwlljwt@163.com; 2School of Physics and Electronics, Shandong Normal University, Jinan 250014, China; w851839145@163.com; 3School of Computer Science & Technology, Shandong University of Finance and Economics, Jinan 250014, China; 19862127463@163.com

**Keywords:** video face tracking, particle filter (PF), features fusion, updating model, template drift

## Abstract

With the revolutionary development of cloud computing and internet of things, the integration and utilization of “big data” resources is a hot topic of the artificial intelligence research. Face recognition technology information has the advantages of being non-replicable, non-stealing, simple and intuitive. Video face tracking in the context of big data has become an important research hotspot in the field of information security. In this paper, a multi-feature fusion adaptive adjustment target tracking window and an adaptive update template particle filter tracking framework algorithm are proposed. Firstly, the skin color and edge features of the face are extracted in the video sequence. The weighted color histogram are extracted which describes the face features. Then we use the integral histogram method to simplify the histogram calculation of the particles. Finally, according to the change of the average distance, the tracking window is adjusted to accurately track the tracking object. At the same time, the algorithm can adaptively update the tracking template which improves the accuracy and accuracy of the tracking. The experimental results show that the proposed method improves the tracking effect and has strong robustness in complex backgrounds such as skin color, illumination changes and face occlusion.

## 1. Introduction

Target tracking technology is an important computer vision research field, which is widely used in the Internet of Things and artificial intelligence [1,2]. Face recognition technology can effectively realize real-time multi-objective online retrieval and comparison in crowded areas such as banks, and the actual application effect is good. Moreover, face information [3] is easy to collect, difficult to copy and steal, and natural and intuitive. Therefore, face recognition technology has become the preferred choice for commercial banks’ security prevention and control measures. Many face tracking methods have been proposed in the past few years. At present, two excellent algorithms, namely mean shift [4] and particle filter [5], have been widely used for target tracking. The mean shift algorithm is a kind of deterministic tracking algorithm, which looks for the nearest pattern of points in the sample distribution. The particle filter algorithm is a Monte Carlo simulation method based on non-parametric to achieve recursive Bayesian estimation algorithm, which can effectively solve nonlinear and non-Gaussian state estimation problems. The most popular target tracking cues include color feature [6], edge feature [7], texture feature [8] and motion feature [9]. Each feature has its own advantages and disadvantages in the application. For example, the tracking method based on color features is insensitive to the rotation and posture of the face in which the face can be tracked in real time. However, when the illumination changes and the skin-like object appears [10] in the scene, it is difficult to accurately track the face, so the face tracking based on a single clue is not robust. Therefore, more and more researchers combine multiple features to improve tracking performance. Particle filter-based tracking algorithms are becoming more and more popular because of their high tracking accuracy and strong anti-interference [11].

The aim of this study is the tracking of a single face in a video in which the images are often affected by complex backgrounds such as illumination change, face rotation and scale change [12], and face occlusion. The adaptive fusion of color and edge features reduces the complexity of the environment. The particle filter target tracking window is adaptively adjusted [13] and the template is adaptively updated [14] in the face tracking system. In addition, the particle filter algorithm together with the histogram algorithm improve the speed of the algorithm [15]. The improved tracking algorithm is capable of adaptively adjusting the [16] tracking window scale to obtain stable tracking of objects with significant dimensional changes. In the tracking process, the template is adaptively updated with the change of the object [17]. The experimental results show that the algorithm has anti-background interference ability, good stability and robustness [18], and finally realizes accurate real-time tracking of the face.

## 2. Particle Filter Algorithm

The particle filter tracking algorithm is a Bayesian recursive estimation algorithm whose purpose is to construct the posterior probability distribution of the target state. The essence of Bayesian recursive filtering is to use the prior knowledge to construct the posterior probability density function of the random state of the target system, and use some estimation criterion to estimate the state value of the target. When the mean square error of this algorithm is the smallest, it can be considered to be optimal. It is generally assumed that the dynamic model has a first-order Markov state transition property, and the observations are conditionally independent with respect to the state [19]. The initial state probability density function of the target system is assumed to be known, where the initial state vector of the target system the initial information of the system measurement value is expressed. The System State Space Model (SSD: State Space Model) is established as follows:(1)$$x_{k}=f_{k}\left(x_{k-1},v_{k}\right)$$
(2)$$y_{k}=h_{k}\left(x_{k},n_{k}\right)$$

Among them, *h*(.) *f*(.) are the target state transfer function and the observation function observation equation *ν_k_ n_k_* system noise and observed noise, respectively.

In order to obtain the exact solution of the posterior probability distribution *p*(*x_k_*|*y*_1:*k*_) of the target in each frame, it is necessary to predict and update the target. The target state can be estimated after repeated iterations:(3)$$p\left(x_{k}|y_{1:k-1}\right)=\int p\left(x_{k}|x_{k-1}\right)p\left(x_{k-1}|y_{1:k-1}\right)dx_{k-1}\;p\left(x_{k}|y_{1:k}\right)=\frac{p\left(y_{k}|x_{k}\right)p\left(x_{k}|y_{1:k-1}\right)}{p\left(y_{k}|y_{1:k-1}\right)}$$

Through the Bayesian recursive filtering process, the target posterior probability density function *p*(*x_k_*|*y*_1:*k*_) to be tracked can be obtained. Then a state quantity can be optimally estimated as the current tracking result of the target object. The target state estimation result is:(4)$${\overset{\wedge }{x}}_{k}=\int x_{k}p\left(x_{k}|y_{1:k}\right)dx_{k}$$

When Bayesian recursive filtering is used to solve the posterior probability density *p*(*x_k_*|*y*_1:*k*_) of the target state, there is integral operation in the formula, which cannot be completed analytically. For a general nonlinear non-Gaussian system, it is very difficult to solve the high-dimensional variables. Therefore, a suboptimal solution under the Bayesian framework is used:(5)$$\underset{N\to \infty }{\lim}\overset{\wedge }{E}\left[f\left(X_{0:k}\right)\right]=E\left[f\left(X_{0:K}\right)\right]$$

### 2.1. Sequential Importance Sampling

Sequential importance sampling avoids the disadvantages of taking too long to run to a stationary state and not being clear when reaching a stationary state. Sequential importance sampling is based on the sequential analysis method in statistics to estimate the posterior probability density. In fact, it is difficult to sample from the posterior probability density function *p*(*x*_0:k_|*y*_1:*k*_) of the real target. Assuming that the importance density function can be decomposed into:(6)$$q\left(x_{0:k}y_{1:k}\right)=q\left(x_{k}|x_{0:k-1},y_{1:k}\right)q\left(x_{0:k-1}|y_{1:k-1}\right)$$

The analytical formula of the posterior probability distribution at the current moment is:(7)$$\begin{array}{ll} p\left(x_{0:k}|y_{1:k}\right) & =\frac{p\left(x_{0:k},y_{k}|y_{1:k-1}\right)}{p\left(y_{k}|y_{1:k-1}\right)}\\ & =\frac{p\left(y_{k}|x_{0:k},y_{1:k-1}\right)p\left(x_{0:k}|y_{1:k-1}\right)}{p\left(y_{k}|y_{1:k-1}\right)}\\ & =\frac{p\left(y_{k}|x_{0:k},y_{1:k-1}\right)p\left(x_{k}|x_{0:k-1},y_{1:k-1}\right)p\left(x_{0:k-1}|y_{1:k-1}\right)}{p\left(y_{k}|y_{1:k-1}\right)} \end{array}$$

According to the Markov nature, the above formula can be written as:(8)$$p\left(x_{0:k}|y_{1:k}\right)=\frac{p\left(y_{k}|x_{k}\right)p\left(x_{k}|x_{k-1}\right)p\left(x_{0:k-1}|y_{1:k-1}\right)}{p\left(y_{k}|y_{1:k-1}\right)}$$

Particle weights $\omega_{k}^{i}$ can be recursively expressed as follows:(9)$$\begin{array}{ll} \omega_{k}^{i} & \propto \frac{p\left(x_{0:k}^{i}|y_{1:k}\right)}{q\left(x_{0:k}^{i}|y_{1:k}\right)}\\ & =\omega_{k-1}^{i}\propto \frac{p\left(y_{k}|x_{k}^{i}\right)p\left(x_{k}^{i}|x_{k-1}^{i}\right)}{q\left(x_{k}^{i}|x_{0:k-1}^{i},y_{1:k}\right)} \end{array}$$

If $q\left(x_{k}|x_{0:k-1}^{i},y_{1:k}\right)$ satisfies the first-order Markowitz conditions: $q\left(x_{k}/x_{0:k-1}^{i},y_{1:k}\right)=q\left(x_{k}|x_{k-1}^{i},y_{k}\right)$ then:(10)$$\omega_{k}^{i}\propto \omega_{k-1}^{i}\frac{p\left(y_{k}|x_{k}^{i}\right)p\left(x_{k}^{i}|x_{k-1}^{i}\right)}{q\left(x_{k}^{i}|x_{k-1}^{i},y_{k}\right)}$$

The posterior probability density function *p*(*x_k_*|*y*_1:*k*_) at the current moment of the target can be approximated as:(11)$$p\left(x_{k}|y_{1:k}\right)\approx \sum_{i=1}^{N}{\tilde{\omega }}_{k}^{i}\delta \left(x_{k}-x_{k}^{i}\right)$$
where *N* is the number of particles, at *N* →∞ that time, the result of this formula is getting closer to the true target state posterior probability density *p*(*x_k_*|*y*_1:*k*_).

### 2.2. Importance Density Function Selection

In the proposed algorithm, the appropriate importance density function $q\left(x_{k}|x_{k-1}^{i},y_{k}\right)$ is used as it is related to the effective sample size of the particle filter. The optimal choice is expressed as follows:(12)$$\begin{array}{ll} q{\left(x_{k}|x_{k-1}^{i},y_{k}\right)}_{opt} & =p\left(x_{k}|x_{k-1}^{i},y_{k}\right)\\ & =\frac{p\left(y_{k}|x_{k}^{i},x_{k-1}^{i}\right)p\left(x_{k}^{i}|x_{k-1}^{i}\right)}{p\left(y_{k}|x_{k-1}^{i}\right)}\\ & =\frac{p\left(y_{k}|x_{k}^{i}\right)p\left(x_{k}^{i}|x_{k-1}^{i}\right)}{p\left(y_{k}|x_{k-1}^{i}\right)} \end{array}$$

Substituting (13) into (14) gives the updated particle weights:(13)$$\omega_{k}^{i}=\omega_{k-1}^{i}p\left(y_{k}|x_{k-1}^{i}\right)=\omega_{k-1}^{i}\int p\left(y_{k}|x_{k}^{i}\right)p\left(x_{k}^{i}|x_{k-1}^{i}\right)dx_{k}^{i}$$

From the above derivation, it can be seen that the probability density function of optimal importance needs to be sampled from $q\left(x_{k}|x_{k-1}^{i},y_{k}\right)$ and every new state has to be integrated, In order to obtain the recommended distribution simply and effectively, the prior probability density function is often taken as the importance density function, namely:(14)$$q\left(x_{k}^{i}|x_{k-1}^{i},y_{k}\right)=p\left(x_{k}^{i}|x_{k-1}^{i}\right)$$

The particle weight is updated by:(15)$$\omega_{k}^{i}=\omega_{k-1}^{i}p\left(y_{k}|x_{k}^{i}\right)$$

### 2.3. Resampling Technique

The importance sampling algorithm is prone to particle degradation in practical applications [20]. The main reason is that with the process of iteration, the weight of most particles becomes very small or even zero, and a large amount of time will be wasted in the calculation of small weight particles, and the variance of the importance weight will gradually increase, resulting in the error of the posterior probability density function of the target will also increase. Researchers have tried to increase the number of samples *N*, but the effect is unsatisfactory. At present, the commonly used solutions are: (1) choosing a good recommendation distribution; (2) using re-sampling technology [21,22].

In order to accurately measure the degree of particle weight degradation, the concept of effective sample size *N_eff_* is proposed:(16)$$N_{eff}=\frac{N}{1+\mathrm{var}\left(\omega_{k}^{i}\right)}\approx \frac{1}{\sum_{i=1}^{N}{\left(\omega_{k}^{i}\right)}^{2}}$$

From the above formula, it can be seen that the larger the particle weight, the smaller the number of effective particles, and the more serious the degradation of the particle weight. Scholars have studied many re-sampling strategies, such as uniform sampling, Markov chain Monte Carlo (MCMC) moving algorithm, hierarchical sampling [23], and evolutionary algorithm [24]. Among them, the uniform sampling method is the most widely used, that is, the particle weight is 1/*N*. Because of the continuous replication of large-weight particles, resulting in the lack of diversity of particles, not all States of particles need to be re-sampled. A threshold *N_th_* (usually 2/3*N*) is set in advance. When *N_eff_* < *N_th_*, the algorithm is re-sampled, otherwise it will not be re-sampled.

## 3. Results Particle Filter with Multi Features Fusion

In particle filter, observation is based on features. This section will mainly describe the human face features, which are utilized in tracking human face of interest by combining color feature and edge feature.

### 3.1. Color Feature Description

Color histograms [25] have been widely applied because of their insensitivity of rotation and scales. Meanwhile, its calculation is simple. The difference of face color is mainly reflected in the brightness, whereas the hue is uniform, so this paper employs the hue-saturation-value (HSV) color model. Furthermore, this paper only calculates H-S histogram using 16 × 8 bins. The color histogram is shown in Figure 1b. To a certain extent, this model reduces the effect of illumination variation.

The weighted color histogram [26] is constructed as:(17)$$p_{c}\left(n\right)=C_{c}\sum_{i=1}^{N}K_{E}\left(\frac{x_{i}-x_{0}}{a}\right)\delta \left[h\left(x_{i}\right)-n\right]$$
where the pixel *n* ∈ [1, *M*] and *M* is the total number of color weighted histogram bin, the function *h*(*x_i_*) maps the pixel *x_i_* location to the corresponding histogram bin, *x*_0_ is the center position of observation area, and *K_E_*(.)represents the Epanechnikov kernel profile with the nuclear bandwidth *a*.

The Epanechnikov kernel is described as:(18)$$K_{E}\left(x\right)=\left\{ \begin{array}{ll} c\left(1-{\left\|x\right\|}^{2}\right) & \left\|x\right\|<1\\ 0 & \left\|x\right\|\geq 1 \end{array}\right.$$
where *δ* denotes the Kroenke delta function and *C_c_* is a normalization constant:(19)$$C_{c}=\frac{1}{\sum_{i=1}^{N}K_{E}\left(\frac{x_{i}-x_{0}}{a}\right)}$$

The distance between reference target template *q_c_*(*n*) and candidate target template *p_c_*(*n*) can be measured by the Bhattacharyya distance *d*:(20)$$d_{c}=\sqrt{1-\rho_{c}\left[p_{c}\left(n\right),q_{c}\left(n\right)\right]}$$
where *ρ_c_*[*p_c_*(*n*), *q_c_*(*n*)] is the Bhattacharyya coefficient [23]:(21)$$\rho_{c}\left[p_{c}\left(n\right),q_{c}\left(n\right)\right]=\sum_{n=1}^{M}\sqrt{p_{c}\left(n\right)q_{c}\left(n\right)}$$

Face color likelihood function is defined as:(22)$$p\left(y_{c}|x\right)=\frac{1}{\sqrt{2\pi \sigma_{c}}}\exp\left(-\frac{1-\rho_{c}}{2\sigma_{c}{}^{2}}\right)$$
where *σ_c_* is the Gaussian variance which is selected as 0.2.

### 3.2. Edge Feature Description

Edge feature is another key feature of human face. It is insensitive to illumination variation and the background of similar face color. This paper employs edge orientation histogram [24] to depict face edge feature. Meanwhile, we adopt Sobel operator to detect edge contour. The edge orientation histogram is shown in Figure 1c. Firstly, the RGB image was converted to the gray image. Then, we calculate the gradient G and orientation *τ* of each pixel in the target region of each particle as follows:(23)$$p\left(y_{e}|x\right)=\frac{1}{{\sqrt{2\pi \sigma }}_{e}}\exp\left(-\frac{1-\rho_{e}}{2\sigma_{e}{}^{2}}\right)$$
where *G_x_* and *G_y_* represent the level and vertical kernel of images respectively, as well as 0 ≤ *τ* ≤ *π*. Then the edge orientation histogram was constructed as:(24)$$p_{e}\left(n\right)=C_{e}\sum_{i=1}^{N}K_{E}\left(\frac{x_{i}-x_{0}}{a}\right)G\left(x_{i}\right)\delta \left[h\left(x_{i}\right)-n\right]$$

Face edge likelihood function is defined as:(25)$$\left\{ \begin{array}{l} G=\sqrt{G_{x}{}^{2}+G_{y}^{2}}\\ \tau =\arctan\left(\frac{G_{y}}{G_{x}}\right) \end{array}\right.$$
where *σ_e_* is the Gaussian variance and is selected as 0.3, and $\rho_{e}=\;\sum_{n=1}^{M}\sqrt{p_{e}\left(n\right)q_{e}\left(n\right)}$.

### 3.3. Features Fusion Strategy

The observation model *p*(*y_t_*|*x*_t_) in this paper concerns two image cues: color feature and edge feature. According to liner fusion method, the entire observation likelihood function can be calculated as:(26)$$p\left(y_{t}|x_{t}\right)=\theta_{c}p\left(y_{c}|x\right)+\theta_{e}p\left(y_{e}|x\right)$$
where *p*(*y_c_*|*x*) and *p*(*y_e_*|*x*) are the likelihood function of the color feature and edge feature respectively. The terms *θ_c_* and *θ_e_*(0 ≤ *θ_c_*, *θ_e_* ≤ 1) are the weights of the color feature and edge feature, where *θ_c_* + *θ_e_* =≤ 1. The weights in most algorithms are assumed to be unchanged during the tracking, and *θ_c_* = *θ_e_* = 0.5. This is an equal method, whereas the contribution of each feature is different in the actual video tracking. In this paper, we proposed a strategy of self-adaptive multi-features fusion strategy, so that it can make up for the deficiency of each characteristic.

For *s* = {*c*,*e*} its likelihood observation result is $p_{s}^{i}$. The characteristic likelihood value formation observation peak is expressed as:(27)$$\delta_{x}=\frac{1}{N}\sum_{i=1}^{N}\left|p_{x}^{i}-\bar{p_{x}}\right|$$
when ${\bar{p}}_{s}=\frac{1}{N}\sum_{i=1}^{N}p_{s}^{i}$, the value of Equation (17) is as big as possible.

Feature weight is:(28)$$\tilde{\theta_{s}}=\frac{\delta_{s}}{\left|x_{s}^{peak}-{\bar{x}}^{peak}\right|}$$

The weight of each feature can be normalized as:(29)$$\theta_{s}=\frac{\tilde{\theta_{s}}}{\sum_{s=1}^{M}\tilde{\theta_{s}}}$$

Particle weight is updated as:(30)$$w_{t}^{i}\propto w_{t-1}^{i}p\left(y_{t}|x_{t}^{i}\right)=w_{t-1}^{i}\left[\theta_{c}p\left(y_{c}|x\right)+\theta_{e}p\left(y_{e}|x\right)\right]$$

## 4. Face-Tracking System

In this section, the proposed face-tracking system is implemented in detail.

### 4.1. Dynamic Model

The auto-regressive process (ARP) model [19] has been widely used for the purpose of formulating such a dynamical model. The ARP is divided into first-order and second-order. The first-order ARP only considers the displacement and noise of the target. However, moving targets usually have physical properties of velocity and acceleration. Hence, the second-order ARP is adopted in this work. The dynamical model can be represented as:(31)$$X_{t}=AX_{t-1}+BX_{t-2}+CN_{t-1}$$
where A and B are the drift coefficient matrix, C is the random diffusion coefficient matrix, *N_t_*_−1_ is the noise matrix at time (*t*−1). These parameters can be obtained from experience or video sequence training.

### 4.2. Face Tracking with Self-Updating Tracking Window

When the target area of the moving target and the number of particles in the sample are determined, the average distance of the weighted particles in the moving target area to the moving target center is related to the size of the moving target. When the size of the moving target is diminished, the distribution of particles is relatively concentrated, so the average distance of the weighted particles in the moving target area to the moving target center is diminished.

The tracking window size will be smaller or larger than the face size which needs adjustment as the face size changes [9]. The self-updating tracking window model is represented as:
(32)$$\left\{ \begin{array}{l} s_{t}\left(x\right)=s_{t-1}\left(x\right)\times d/d_{l}\\ s_{t}\left(y\right)=s_{t-1}\left(y\right)\times d/d_{l} \end{array}\right.$$
where *s_t_* represents the current frame tracking window size, *s_t_*_−1_ represents the previous frame tracking window, and *d*_1_ is the average distance from the previous frame to the target center.

We define the state variable (*x,y*) as the center of the rectangle. The center of the particle is (*x_i_,y_i_*), and *ω^i^*(*i* = 1, 2, …M) are the weights of the samples. M is the number of particles whose weight is greater than a certain threshold, T. The average distance between the particle and the target center is defined as:(33)$$d=\frac{1}{M}\sum_{i=1}^{M}\sqrt{{\left(x-x_{i}\right)}^{2}+{\left(y-y_{i}\right)}^{2}}$$

### 4.3. Updating Model

In the actual process of human face tracking, the environment or the target may be under various conditions (such as illumination variation, posture variation, and occlusion). A single fixed target model would not be stable for a long time and is prone to drift. In this paper, we present a template updating technology [10]. Threshold value is set at *π*. If the average status *P* < *π* the template updates as:(34)$$H_{\mathrm{new}}=\tau H_{old}+\left(1-\tau \right)H_{current}$$
where *H_new_* is the new reference histogram, *H_old_* is the initial reference histogram, *H_current_* is the current reference histogram. If *π* = 0.3 and *τ* is a constant 0 ≤ *τ* ≤ 1, then *τ* = *ρ^k^*^−1^/*ρ^k^*^−1^ + *ρ^k^*.

### 4.4. The Integral Histogram of the Image

During the process of particle filtering, the purpose is mainly to calculate the similarity coefficient of the histogram of the target and the histogram of all particles, and then determine the observed value of the target. This process is an exhaustive search process. If there are more particles in a video, it will take a long time in calculation [26]. In order to solve this problem, the approach of integrating histogram was used to simplify the calculation of particles histogram, which greatly saved the computation time. The methodology of this method is to simply add and subtract the upper left corner, the upper right corner, the lower left corner and the lower right corner of the particle. Finally, we can get the histogram of the whole region.

### 4.5. Tracking Algorithm Procedure

The specific steps of this algorithm are summarized as follows:

Step 1: Initialize *t* = 0, the known initial state is ${\left\{x_{0},\omega_{0}\right\}}_{i=1}^{N}$, and calculate the template color histogram, edge orientation histogram $p_{c}^{0}$ and $p_{e}^{0}$.

Step 2: Predict the face current state according to Equation (32).

Step 3: Weight updating:
3.1Sample particles $x_{t}^{i}$ from *p*($x_{t}^{i}|x_{t-1}^{i})$, *i* = 1, ···, N;3.2Calculate the color likelihood *p*(*y_c_*|*x*) according to Equation (22);3.3Calculate the gradient likelihood *p*(*y_e_*|*x*) according to Equation (25);3.4Calculate the weight of each feature according to Equation (27), and normalize *θ_c_*,*θ_e_* by Equation (18);3.5Calculate the entire observation likelihood *p*(*y_t_*|*x*) according to Equation (26);3.6Calculate the weight $w_{t}^{.}$ according to Equation (30);3.7Normalize weight ${\tilde{\omega }}_{t}^{i}=\frac{\omega_{t}^{i}}{\sum_{i=1}^{N}\omega_{t}^{i}}$

Step 4: Output target state estimation

Step 5: The particle resamples: $E\left(x_{t}\right)=\;\sum_{i=1}^{N}{\tilde{\omega }}_{t}^{i}x_{t}^{i}$

Step 6: Update the model according to Equation (31).

Step 7: Continue to track and return to Step 2.

## 5. Experimental Results and Analysis

In this section, we will test the effectiveness and accuracy of our proposed algorithm. The data set was the Visual Tracker Benchmark. This benchmark includes the results from 100 test sequences and 29 trackers. This website contains data and code of the benchmark evaluation of online visual tracking algorithms. Visual Tracker Benchmark Database is applied in experiment which provides a standard video database for evaluating face tracking and recognition algorithms.

### 5.1. Tracking Effect and Error Analysis

In order to assess the tracking performance of this proposed method, we manually selected a rectangle window of face as the matching template, and set N = 100. This experiment platform is based on Visual Studio 2010 OpenCV 2.4.8. Euclidean distance is used to measure the result of target tracking. The sequences are obtained from reference [27], and the information of them is shown in Table 1.

We carried out experiments on different video sequences. In the framework of the improved particle filter algorithm, we extracted face color features and edge features. The adaptive fusion strategy proposed in this paper was used to track video faces. At the same time, the test results of this algorithm were compared with the tracking results of the traditional particle filter based on the color feature, so as to verify the advantages of our improved tracker in the presence of occlusion, similar background and illumination changes.

Furthermore, Root Mean Squared Error (RMSE) is used as one objective metrics to evaluate the quantitative quality of different experimental results. The smaller the RMSE, the closer the tracking result is to the ground truth.

The first sequence in Figure 2 is about drastic variations in illumination, where the cast shadow drastically change the appearance of the target face when a person walks underneath a trellis covered by vines. The frame indexes are the 24th, 36th, 52th and 138th, respectively, shown as Figure 2, where the blue points denote the particles and red boxes are the tracking results. Figure 2a is the tracking result based only on color features. In the tracking process, the performance is ideal for the condition that the outdoor illumination variation is not obvious. When the light of target tracking based on color feature changes dramatically, the rectangular box cannot select the correct face location. This happens because the color features are very sensitive to illumination variation. Figure 2b shows the tracking results of this proposed algorithm. At the 36th frame and 138th frame, compared to the undesirable results based on color feature, our algorithm overcomes the problems and tracks the face during the entire challenging sequence successfully. Figure 3 is the quantitative analysis of the tracking results measuring the position error, which indicates that the error rate of our algorithm is obviously lower than single color feature tracking. This is mainly attributed to the full use of the latest observation information in the improved particle filtering algorithm and the fusion strategy of two features, so that the target can be tracked more accurately.

Figure 4 allows us to test the robustness of the experiment. It can be seen from the test results that the tracking effect of the algorithm is very stable.

The second sequence in Figure 5 is about the variations in pose and background, corresponding to the 6th, 38th, 82th, 117th frame snapshots. Figure 5 shows the comparison results between our algorithm and the use of color features. If we only use color features for a particle filter, the tracking algorithm is not sensitive to the changes of facial expression and posture. However, when there are skin-like objects in the video, the algorithm considers the object which is similar to the human face as the face area and causes a tracking failure, as shown in Figure 5a. For example, in the 82th frame, the human face is interfered by hands, which leads to the tracking deviating from the real face position and tracking to the hand. In comparison, our method can track the face very well, as shown in Figure 5b. Furthermore, we can see from Figure 6 that the error of our algorithm is very small, conforming to the observation of Figure 5. Figure 7 allows us to test the robustness of the experiment. It can be seen from the test results that the tracking effect of the algorithm is very stable.

The third sequence in Figure 8 is about enlarged objects and diminished objects. The size of the video sequence is 480 × 360 pixels. The frame indices are 8, 16, 19, and 40. A girl in the computer room (a frame with a simpler background) is used to evaluate the performance of the proposed algorithm in handling pose variation. The results are shown in Figure 8. This is a stretched video shot where the scale of the human face is variable. Scale variation occurs when the girl goes toward the camera, as seen from Figure 6. Figure 8a shows results of the skin color tracker. The red rectangle is always the same size throughout the entire process, but the face partially goes outside the red rectangle (such as the 16th frame and the 19th frame). Then, the girl slowly turns away from the camera, and the red rectangle contains other information except for the human face. The algorithm excellently scaled the size of the tracking window to the human face as it changed. As shown in Figure 8b, it mainly benefits from adaptively adjusting the tracking window in the tracking algorithm.Figure 9 gives the error curves comparison about Test 3. The tracking error of the algorithm our proposed is obviously lower than that based on color feature.

The fourth sequence in Figure 10 is about the rotation of a man’s head. The frame indices are 6, 27, 33, and 41. Some samples of the final tracking results are shown in Figure 10. When the man turned his head, his head looks changed. The result of Figure 10a benefits from the color feature. In the 27th frame, the man’s face drastically changes, and the target is lost in the succeeding frames. Especially, as shown in the 33th frame, the overlarge tracking window has made the tracker lose the target. The results of the tracking algorithm proposed in this paper are shown in Figure 10b. This method is adaptive template updating, which accurately tracks the target and is robust to pose variations. Based on Table 1, the tracking performance of our method is the best. Figure 11gives the error curves comparison about Test4. The tracking error of the algorithm our proposed is obviously lower than that based on color feature.

The fifth sequence in Figure 12 is about the face blocked by a book. Some samples, corresponding to the 13th, 167th, 268th and 278th frame snapshots, of the final tracking results are shown in Figure 12. Performance on this sequence exemplifies the accuracy and robustness of our fusion method to partial occlusion. As we can observe, when the face is occluded, the visible face target range becomes smaller and smaller, and the face information is not comprehensive. At the 167th frame that the man is covered partly by a book, the tracking method based on color feature loses the human face obviously, as shown in Figure 12a. Moreover, when the book is taken away, it still fails to track the man’s face, which confuses the tracker from following the right face within the interruption. In like manner the tracking method based on edge feature loses the human face obviously, as shown in Figure 12b. The tracking results of our method are shown in Figure 12c. Although both methods can track faces, when a face is occluded or the illumination changes, the single feature method cannot track the face in real time. Nevertheless, our fusion tracker follows the man’s face accurately and robustly. Figure 13 gives the error curves comparison about Test5 the tracking error of the algorithm our proposed is obviously lower than that based on colora and egde feature. It is mainly benefited from the proposed fusion scheme, which can enhance the one that is reliable for tracking.

The fourth sequence in Figure 14 is about high-speed face movement and lens stretching. The frame indices are 16, 54, 101 and 118. In this sequence, Figure 14a is based on the algorithm in the literature. When the rapid rotation occurs, the template cannot be updated in time to cause the tracking effect to be extremely poor. When the lens is stretched, the tracking window size cannot be adjusted to cause redundant or missing information. Figure 14b is a comparison of the algorithm in this paper. Our tracker can accurately locate the face even if the face part moves at high speed. The red rectangular frame always surrounds the human face, which can track the face very well. It mainly benefits from the improvement of the particle filter algorithm and the adaptive fusion strategy using two features to make up for the disadvantages of each feature. In addition, the real-time update of the face template and the window adaptive adjustment reduce the impact of the target high-speed motion and stretching. Figure 15 gives the error curves comparison about Test6 the tracking error of the algorithm our proposed is obviously lower than that based on colora and egde feature.

The third sequence in Figure 16 is about light change. The size of the video sequence is 480 × 360 pixels. The frame indices are 12, 22, 34, and 47. When the illumination changes, Figure 16a is a face tracking based on the single feature of color. From the figure we can see that only the single feature of color is easy to cause information loss. When we combine edge features and color features, Through Figure 16b, we can see that the face part can be accurately tracked regardless of the change in illumination. When the illumination changes, Figure 16a is a face tracking based on the single feature of color. From the figure we can see that only the single feature of color is easy to cause information loss. When we combine edge features and color features, Through Figure 16b, we can see that the face part can be accurately tracked regardless of the change in illumination. Figure 17 gives the error curves comparison for Test 7. The tracking error of the algorithm our proposed is obviously lower than that based on color feature.

### 5.2. Comparison with Other Algorithms

In order to further verify the accuracy of our algorithm, we further compare the algorithm of this paper with the orthers algorithm.Figure 18 and Figure 19 are comparisons with the KCF algorithm. They are the quantitative analysis of the tracking results measuring the position error, which indicates that the error rate of our algorithm is obviously lower than with single color feature tracking.

Figure 20 and Figure 21 are comparisons with the Fiserface algorithm. By comparison, we find that the tracking accuracy of our algorithm is higher. It can be seen from the test results that the tracking effect of the algorithm is very stable. Our comparison with the Fiserface algorithm is mainly to compare the accuracy of the tracking effect, and the accuracy is used to show that our algorithm is feasible. Moreover, compared to Fiserface as the number of frames increases, the experimental results obtained by our algorithm are better, and our algorithm requires less computational time and cost under the same situation.

Figure 22 and Figure 23 are comparisons with the R-CNN algorithm. By comparison, we find that the tracking accuracy of our algorithm is higher. Our comparison with the R-CNN algorithm is mainly to compare the accuracy of the tracking effect and the accuracy is used to show that our algorithm is feasible. Moreover, compared to R-CNN as the number of frames increases, the experimental results obtained by our algorithm are better, and our algorithm requires less computational time and cost under the same situation.

### 5.3. Computational Efficiency

In order to verify the calculation efficiency of integral histogram is higher than normal histogram, two histograms of video face tracking were used to compare their consumption in the process of tracking in the histogram computation time.

Table 2 shows that with the increase of particle number, computational time is also gradually increased, but the time consumption growth speed of the integral histogram is not bigger than for a normal histogram. Also, we found that in the case of few particles, the ordinary histogram calculation time is less than the integral histogram, because the initialization time of the integral histogram is basically fixed. Therefore, more particles exist in a video, more obvious the advantage of the integral histogram.

## 6. Conclusions

In this paper, a face tracking algorithm based on adaptive fusion of skin color and edge features is proposed, which adaptively updates the template and adaptively adjusts the target tracking window to adapt to complex video background. Our experimental results show the advantages of our algorithm more clearly. It can be seen from the table in Figure 2 that our algorithm takes less time and is more efficient. The experimental results show that compared with the single feature algorithm, the face can be accurately tracked in complex backgrounds with skin color, illumination changes, and especially in face color changes. The algorithm reduces the loss of target information by updating the tracking target template in real time, which further improves the accuracy of the algorithm. In the future, the computational complexity of particle filter algorithm will be thoroughly studied to meet the real-time requirements.

However, there are some limitations of the proposed algorithm. For example, the hardware- assisted approach need to be considered, the initial face template should be well defined, or the target model gets anchored to the first frame. In our future works, we plan to develop a faster and more robust tracking method.

## Figures and Tables

**Figure 1 sensors-19-01245-f001:**
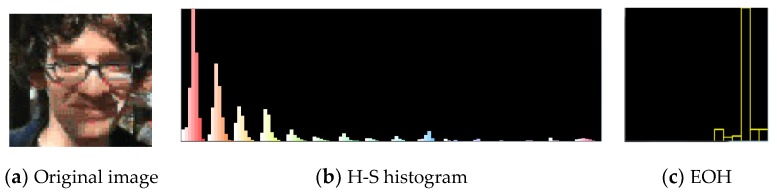
Color histogram and edge orientation histogram.

**Figure 2 sensors-19-01245-f002:**
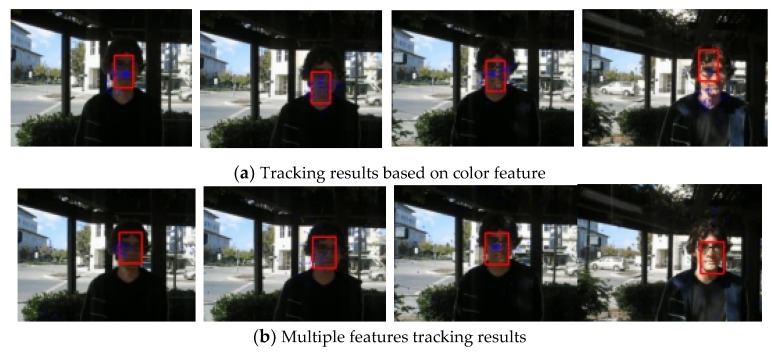
Tracking results under the illumination variation.

**Figure 3 sensors-19-01245-f003:**
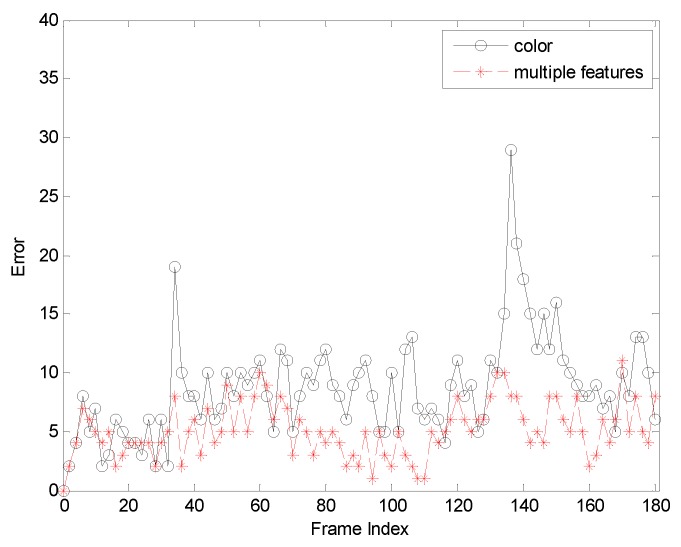
The RMSE comparison corresponding to illumination variation.

**Figure 4 sensors-19-01245-f004:**
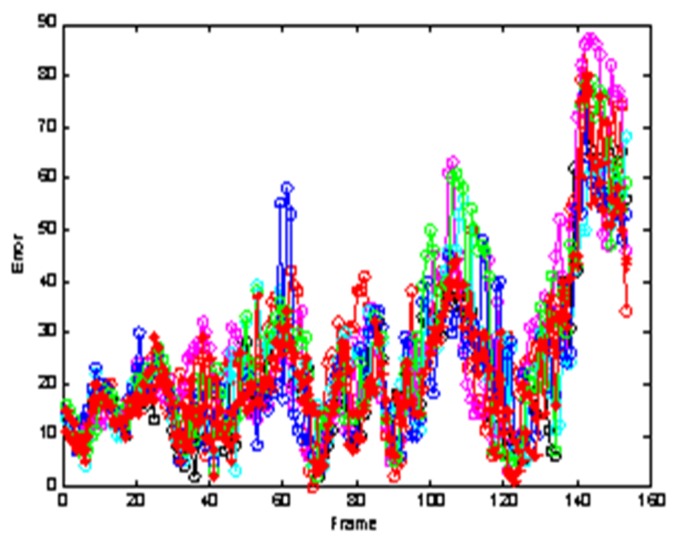
The analysis robustness.

**Figure 5 sensors-19-01245-f005:**
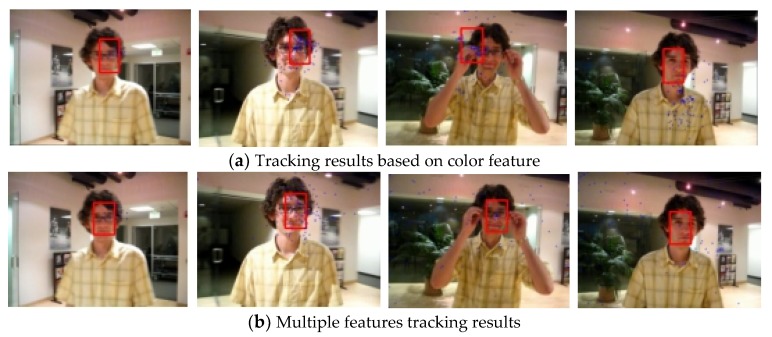
Tracking results under the background changes.

**Figure 6 sensors-19-01245-f006:**
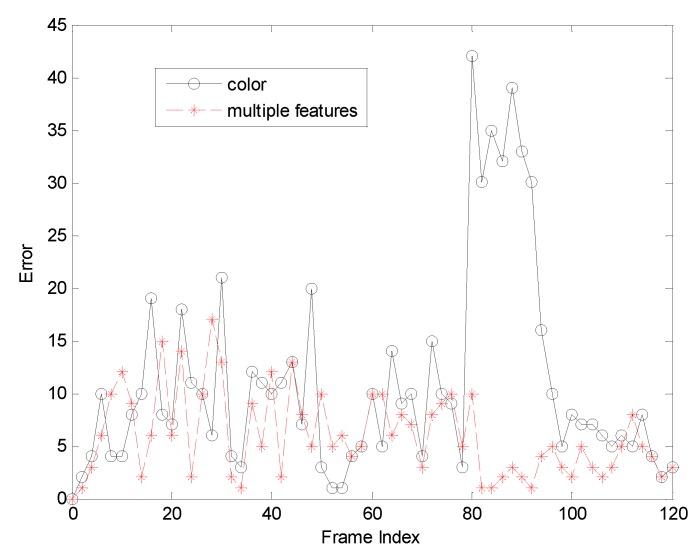
The RMSE comparison corresponding to background changes.

**Figure 7 sensors-19-01245-f007:**
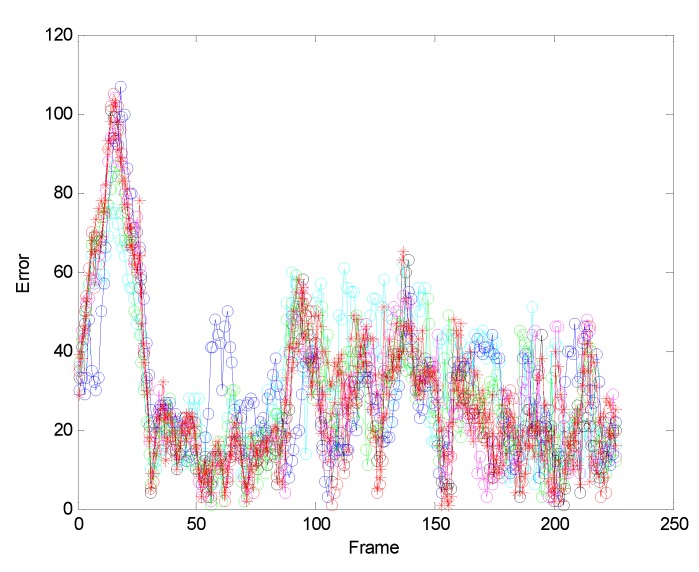
The analysis robustness.

**Figure 8 sensors-19-01245-f008:**
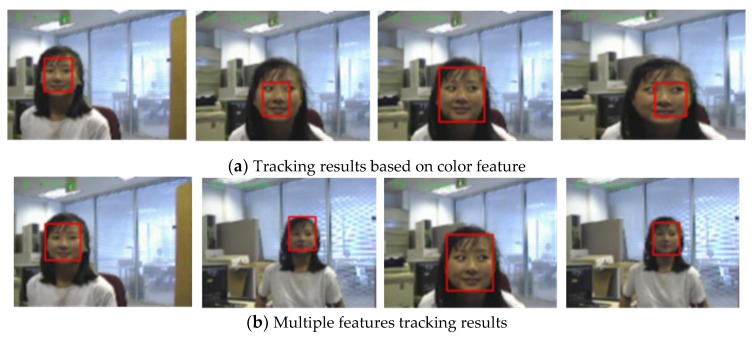
Tracking results under the background changes.

**Figure 9 sensors-19-01245-f009:**
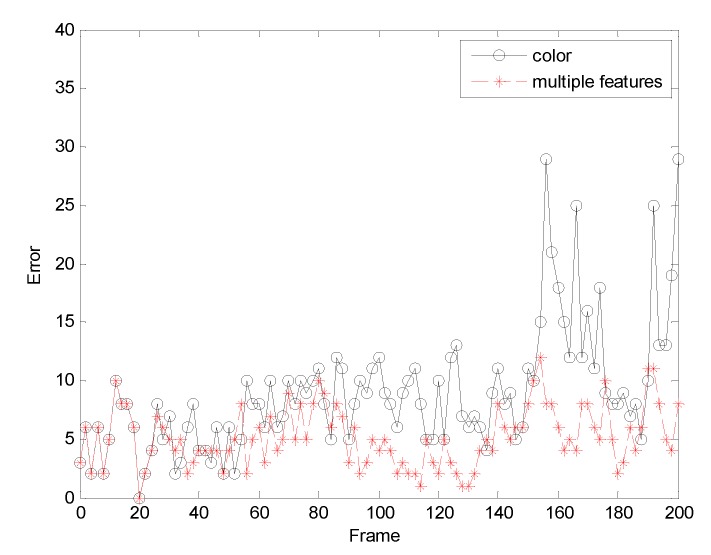
The RMSE comparison corresponding to background change.

**Figure 10 sensors-19-01245-f010:**
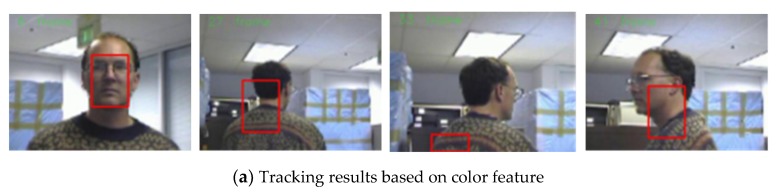
Tracking results under the background changes.

**Figure 11 sensors-19-01245-f011:**
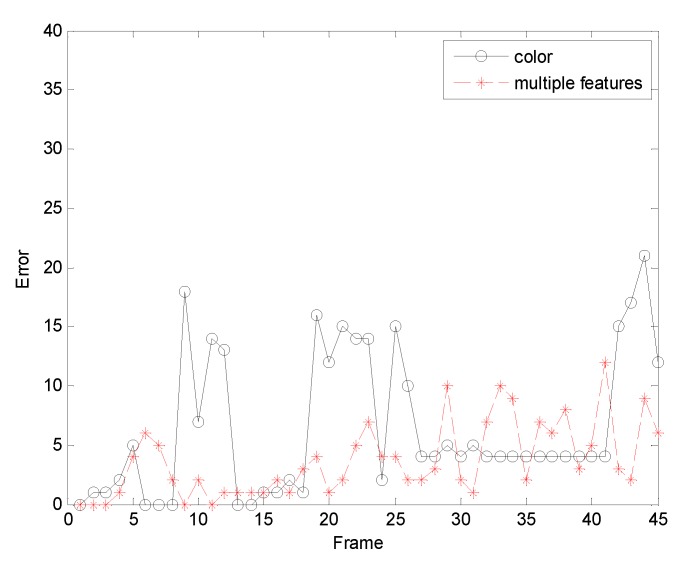
The RMSE comparison corresponding to background changes.

**Figure 12 sensors-19-01245-f012:**
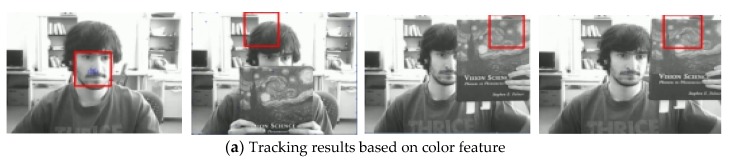
Tracking results under the background changes.

**Figure 13 sensors-19-01245-f013:**
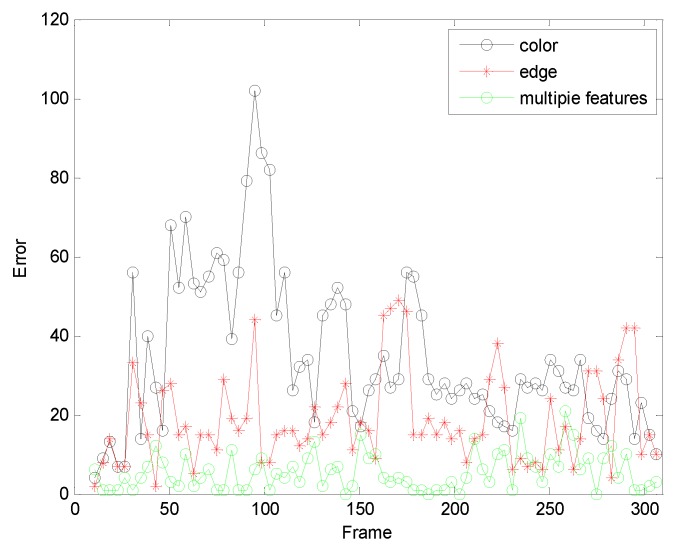
The RMSE comparison corresponding to background changes.

**Figure 14 sensors-19-01245-f014:**
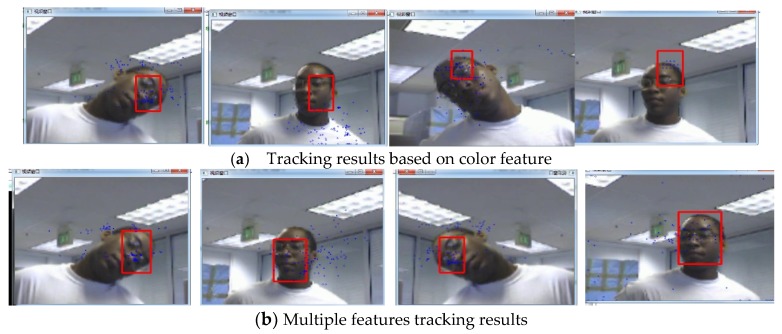
Tracking results under the background changes.

**Figure 15 sensors-19-01245-f015:**
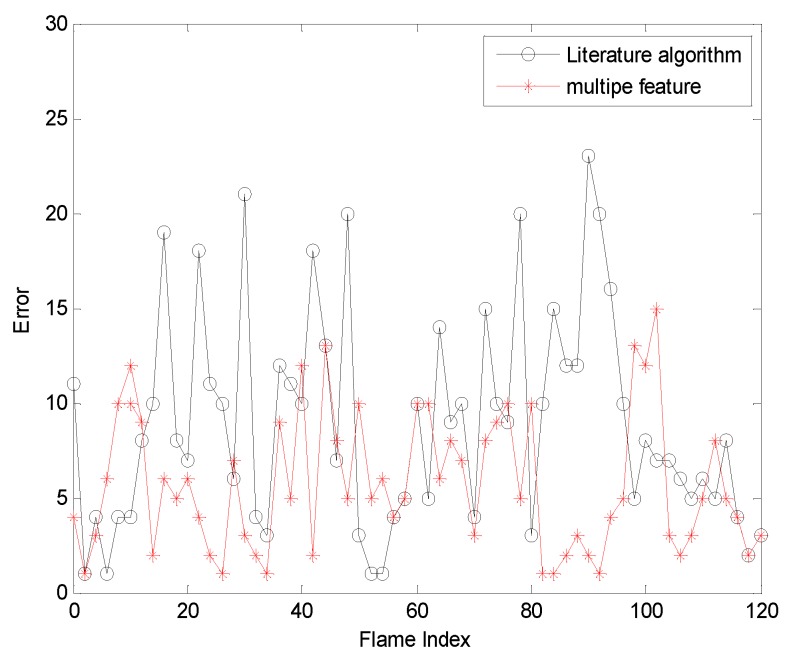
The RMSE comparison corresponding to background changes.

**Figure 16 sensors-19-01245-f016:**
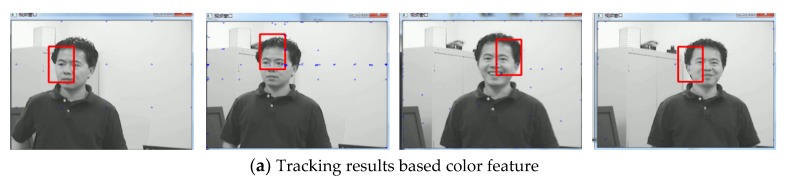
Tracking results under the background changes.

**Figure 17 sensors-19-01245-f017:**
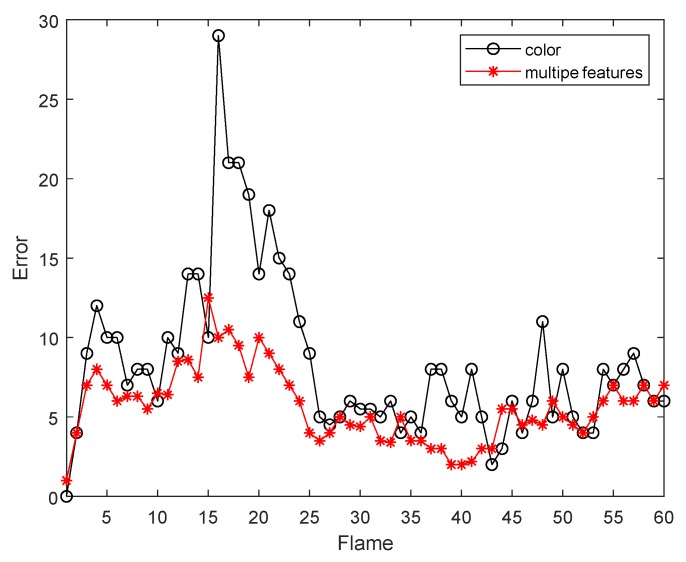
The RMSE comparison corresponding to background change.

**Figure 18 sensors-19-01245-f018:**
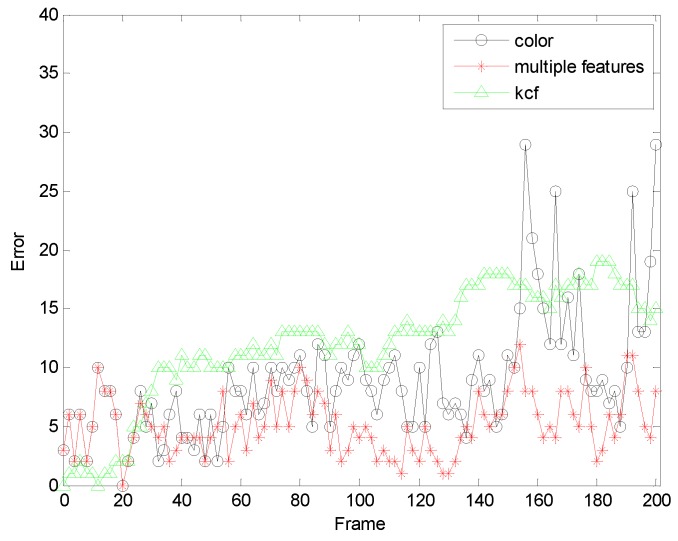
The RMSE comparison corresponding with KCF.

**Figure 19 sensors-19-01245-f019:**
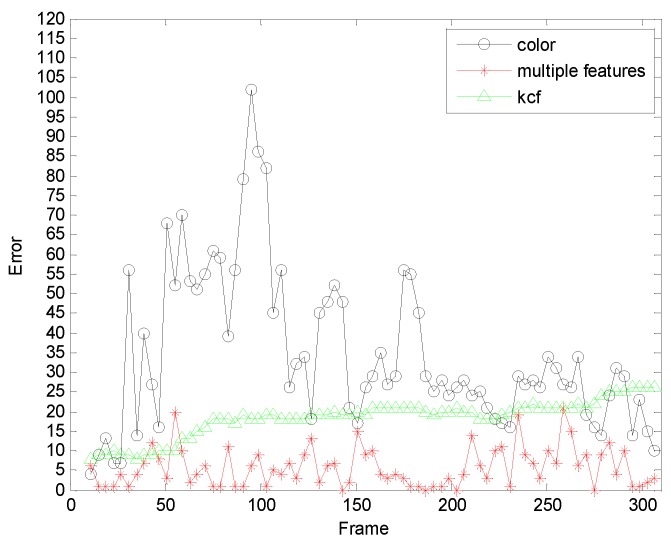
The RMSE comparison corresponding with KCF.

**Figure 20 sensors-19-01245-f020:**
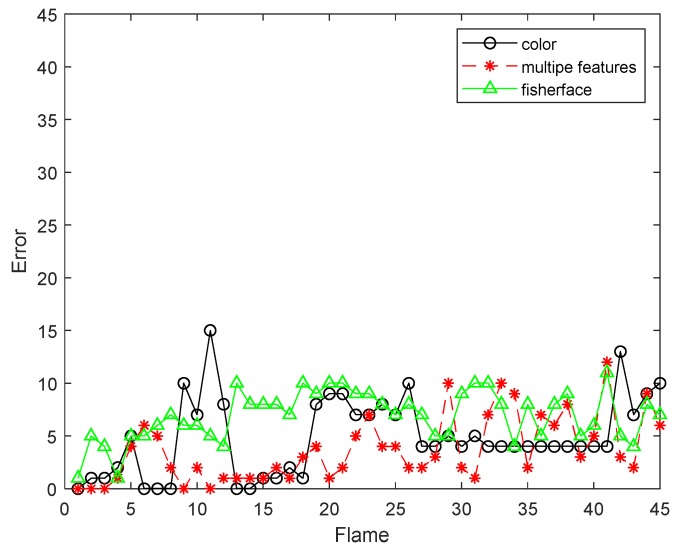
The RMSE comparison corresponding with Fisherface.

**Figure 21 sensors-19-01245-f021:**
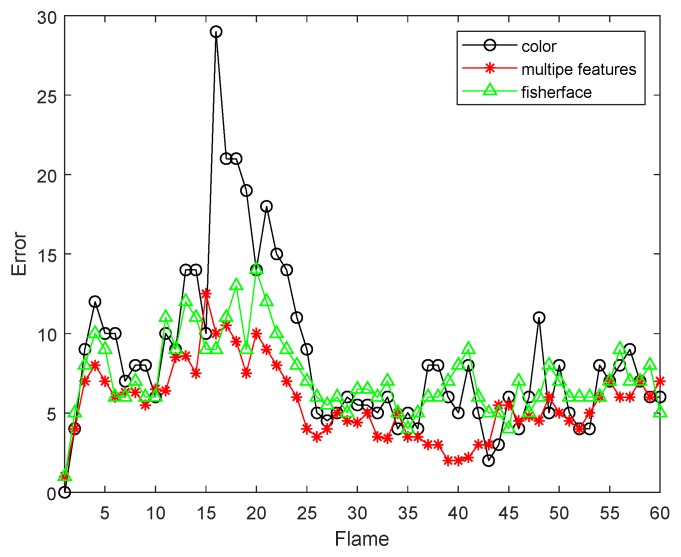
The RMSE comparison corresponding with Fisherface.

**Figure 22 sensors-19-01245-f022:**
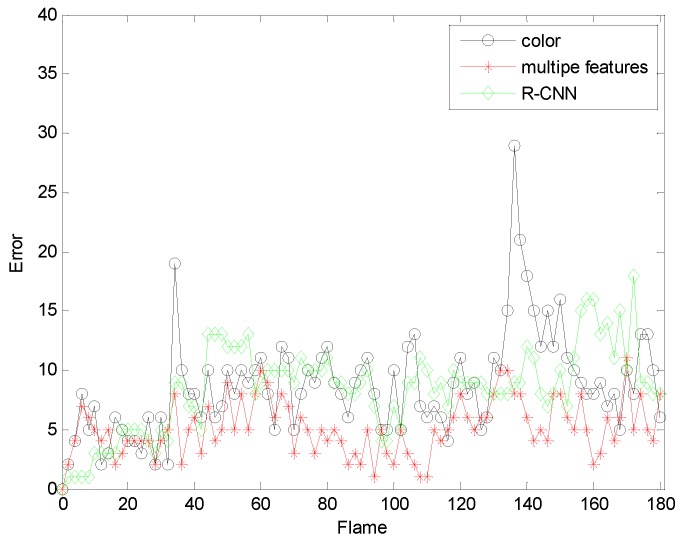
The RMSE comparison corresponding with R-CNN.

**Figure 23 sensors-19-01245-f023:**
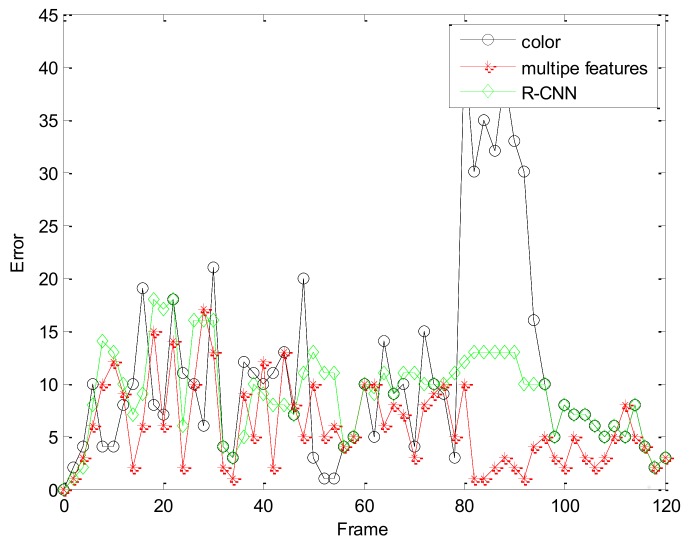
The RMSE comparison corresponding with R-CNN.

**Table 1 sensors-19-01245-t001:** Video Sequences Used in Our Experiments.

Sequence	Frame Size	Sequence Characteristics	Total Frames
**Test 1**	480 × 360	Illumination variation	182
**Test 2**	480 × 360	Similar background	119
**Test 3**	480 × 360	Object scaling	198
**Test 4**	480×360	Object rotation	45
**Test 5**	480 × 360	Occlusion	310
**Test 6**	480 × 360	High speed operation and lens stretching	180
**Test 7**	480 × 360	Light change	60

**Table 2 sensors-19-01245-t002:** The Effects of Different Particle Number on the Calculation.

Particle Number	Time/s	
	Normal Histogram	Integral Histogram
20	0.028987	0.050296
50	0.085672	0.054322
100	0.148562	0.058970
500	0.765326	0.063952

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
