# Peer review of "Research on a Face Real-time Tracking Algorithm Based on Particle Filter Multi-Feature Fusion"

_sensors, 2019, doi:10.3390/s19051245_

Round 1
Reviewer 1 Report
The presentation and language of paper prevent readers to clearly understand the core contribution of the work. i.e., what is new vs. what is already established. The authors mentioned multiple features has already been used for facial recognition in previous works, thus, it is expected to see experiments comparing the authors' algorithms with other multi-feature algorithms. However, the majority of the evaluation are comparing against color-only feature algorithm. Though the improvement is impressive, due to above reasons, it is unclear that how much improvement does the purposed algorithm achieve compare to status quo.
Author Response
Dear editor, thank you very much for your suggestions on my paper. My answer is in all the following documents.

Reviewer 2 Report
This paper presented a particle filter tracking framework updated using the fusion of facial skin color and edge features for face recognition. Experiment results show that the proposed method achieved better results than other existing methods.
However, the novelty of this paper is limited. The multi features are considered in the paper to design particle filter for face tracking, which seems not to be novel. Additionally, what are the advantages of these multi features in the proposed framework? More discussions are needed to show the effectiveness of the proposed framework.
The use of term “multi features” should be justified due to there are only two features used in this paper, i.e., color and edge features.
Moreover, ablation study about color and edge features should be included to validate the proposed algorithm.
Some methods using deformable part models should be compared with the proposed method. For example, Tracking Live Fish From Low-Contrast and Low-Frame-Rate Stereo Videos. IEEE Trans. Circuits Syst. Video Techn. 25(1): 167-179 (2015), and some recent CNN based methods for object detection, i.e., Faster R-CNN, YOLO, and RetinaNet.
The lack of experiments using the metric “Intersection over Union” should be provided.
The writeup is very sloppy and needs being significantly revised.
Author Response

(The authors gave the same response as above.)

Round 2
Reviewer 1 Report
The authors updated intro and added more results in evaluation section. However, the author should still make it clear that which part of the paper describe the novelty content. It currently reads like a survey of a list of existing algorithms.
As for the extra figures added, some of the figure number seems mis-referred in the text, i.e. fig 10/11/12. For comparison against R-CNN, the improvement is hard to tell from the figure itself. Lot of data points seem close to each other between the two algorithms. It will be helpful if the author can quantify the improvement in numbers.
Author Response
Dear editor,
Thank you very much for your comments on our paper. My answer is presented in the following documents. We are looking forward to hearing from you.
kind regards,
Mr.Wang

Reviewer 2 Report
The authors have taken most of my comments. Now the quality of the paper has been improved and my concerns have been cleared.
Author Response
Dear editor,
Thank you very much for your suggestions on our papers during this time. Under your suggestion, now the quality of the paper has been greatly improved.
kind regards,
Mr. Wang